# Cardiorespiratory Fitness: Reference on the Six-Minute Walk Test and Oxygen Consumption in Adolescents from South-Central Chile

**DOI:** 10.3390/ijerph18052474

**Published:** 2021-03-03

**Authors:** Jaime Vásquez-Gómez, Nelson Gatica Salas, Pedro Jiménez Villarroel, Luis Rojas-Araya, Cesar Faundez-Casanova, Marcelo Castillo-Retamal

**Affiliations:** 1Centro de Investigación de Estudios Avanzados del Maule (CIEAM), Universidad Católica del Maule, Talca 3460000, Chile; jvasquez@ucm.cl; 2Magíster en Actividad Física y Deporte, Universidad San Sebastián, Concepción 4030000, Chile; nelson.gaticas@gmail.com (N.G.S.); peedrojv@gmail.com (P.J.V.); 3Instituto Profesional AIEP, Universidad Andrés Bello, Liceo Bicentenario Zapallar de la Municipalidad de Curicó, Curicó 3340000, Chile; frojas.ef@gmail.com; 4Departamento de Ciencias de la Actividad Física, Universidad Católica del Maule, Talca 3460000, Chile; cfaundez@ucm.cl; 5Laboratorio de Rendimiento Humano, Universidad Católica del Maule, Talca 3460000, Chile

**Keywords:** physical fitness, exercise test, oxygen consumption, walking, adolescent

## Abstract

Cardiorespiratory fitness (CRF) provides oxygen to the exercising muscles and is related to body adiposity, with cardiometabolic variables. The aim was to develop reference values and a predictive model of CRF in Chilean adolescents. A total of 741 adolescents of both genders (15.7 years old) participated in a basic anthropometry, performance in the six-minute walk test (SMWT), and in Course Navette was measured. Percentiles were determined for the SMWT, for the V̇O_2_max, and an equation was developed to estimate it. The validity of the equation was checked using distribution assumptions and the Bland–Altman diagram. The STATA v.14 program was used (*p* < 0.05). The 50th percentile values for males and females in the SMWT and in the V̇O_2_max of Course Navette were, respectively, from 607 to 690 and from 630 to 641 m, and from 43.9 to 45 and from 37.5 to 31.5 mlO_2_·kg·min^−1^, for the range of 13 to 17 years. For its part, the model to predict V̇O_2_max incorporated gender, heart rate, height, waist-to-height ratio (WHR), and distance in the SMWT (R^2^ = 0.62; estimation error = 0.38 LO_2_·min^−1^; *p* <0.001). Reference values can guide physical fitness in Chilean adolescents, and V̇O_2_max was possible to predict from morphofunctional variables.

## 1. Introduction

Cardiorespiratory fitness (CRF) is characterized by providing oxygen to generate energy in muscles during exercise; a low level or decreased level of fitness is associated with cardio metabolic disease and mortality [1]. The CRF has been studied in many large-scale population-based investigations considering a large number of participants, wide age ranges, people from different countries, in longitudinal and cross-sectional studies, and performed CRF interaction with other independent variables. In most of these studies, CRF has been evaluated through prediction equations in the absences of the applications of physical stress test and on some occasions, it has been measured by direct method [2,3,4,5,6].

The CRF has been inversely related to body adiposity and cardiometabodic variables in adults, which could improve with an increase in the CRF [7]. These effects could manifest in children and young people since the CRF had also been associated with cardiometabolic health and even mental variables [8]. It is desirable that this population group have adequate CRF levels; however, this variable has decreased in recent years in children and adolescents in Latin America [9]. Furthermore, it has been reported that physical inactivity in Chilean children and adolescents has reached over 80% [10]. When evaluating the CRF in adolescents, field tests were used where performance is estimated through regression equations, and one of the most commonly used tests is the Course Navette test for which various predictive models for adolescents have been developed [11]. Another field test is the six-minute walking test (SMWT); this is a means of easy application, low cost, and is used to evaluate the physical capacity to exercise [12], which has been applied in children and adolescents [13], and various equations to predict distance travelled had also been performed [14].

Until now, current studies are unknown in which the performance in the SMWT in Chilean adolescents and CRF values in this population group are reported. Therefore, the aim was to develop reference values of the distance covered in the SMWT, of CRF in Course Navette, and to generate a predictive model for this last variable for the walk test in Chilean adolescents of both genders in schools from south-central Chile.

## 2. Materials and Methods 

Junior high school students participating (secondary) from south central Chilean schools. An incidental type sample was developed, consisting of 741 students of both genders with an average age of 15.7 years (Table 1). In order to participant, parents or guardians must give their written constant and the students had to sign an agreement. The Scientific Ethics Committee of the Universidad Católica del Maule, Chile (n° 186/2018), approved the present study.

The basic anthropometric variables of body weight, height, waist circumference (WC), waist-to-height ratio (WHR), and body mass index (BMI) were measured. To classify the students according to WC and WHR, normative values of the Ministry of Health of the Government of Chile was used [15], in which a WHR > 0.55 was classified as higher cardiometabolic risk, and the WC was categorized into abdominal obesity, risk of suffering from it, or normal, according to gender and age. For BMI, Food and Nutrition Technical Assistance [16] standards were used according to gender and age.

To evaluate the aerobatic capacity, the SWMT [17] was performed in 30 m long halls, marking the boundaries at its ends, and adolescents were asked to walk as fast as possible without running or performing an air phase during the march; thus, they had to pass back and forth along the hallways. The distance traveled and the perception of effort [18] were recorded after finishing the test, as well as the recovery heart rate through carotid palpation. For the purposes of this study, the distance traveled was classified into the categories of “acceptable” or “needs improvement” according to gender and age, considering the data available in the literature [14].

The CRF was also measured in the Course Navette test in 20-m long hallways, which were marked at the end. An acoustic signal was used to mark the running intensities in which the participants had to give their maximum physical effort, and at the end, the perception of effort was measured [18]. With this, V̇O_2_max (mL·kg·min^−1^) was estimated with a formula proposed by Léger et al. [19] for females and males between 8 and 19 years old, based on the speed in km·h^−1^ of the last bearing completed and age. V̇O_2_max (ml·kg·min^−1^) was classified into the categories “very low,” “moderate,” “high,” and “very high” depending on gender and age considering international references for children and adolescents [1].

Continuous variables were presented as mean values and standard deviations, with the categorical ones on absolute and relative frequencies, plus their respective confidence intervals (CI) to 95% for both cases. Data were compared between males and females using the T-Student test for independent samples or Mann–Whitney U test, when appropriate.

Percentile values and the CI of the CRF were calculated for the distance traveled in the SMWT and V̇O_2_max (ml·kg·min^−1^) in Course Navette according to gender and age. The prevalence of these variables was also determined according to gender, WHR, BMI, and with the Chi square test (x^2^), and the probability of having a higher CRF was estimated by calculating the odds ratio (OR) with 95% CI.

Finally, a multivariate equation was developed to predict the CRF expressed in V̇O_2_max. (L·min^−1^). For this model, the correlation value is estimated, as well as the determination coefficient (R^2^), the standard estimation error, the standardized coefficients, the CI (95%), and the statistical significance of independent variables and the constant. For the validity of the equation, the residual distribution assumptions were checked with the Durbin–Waston test, the normality with the Kolmogorov–Smirnov test, and the homoscedasticity test with graphical representation. Along with this, the degrees of agreement between the V̇O_2_max criterion test and equation developed using the Bland–Altman diagram were verified. All analysis was carried out with the STATA program version 14 and the statistical significance was assumed with a *p*-value < 0.05.

## 3. Results

### 3.1. Sample Adiposity

In Table 1, adolescents’ characteristics are shown. It was observed that there were differences in basic anthropometry, except WHR. Body weight, height, and WC were higher in males and BMI was higher in females. It was also observed that males had higher abdominal obesity and risk of abdominal obesity percentage according to WC, and a higher cardiometabodic risk percentage according to WHR.

### 3.2. Physical Tests

Differences were found between males and females regarding the distance traveled in the SMWT, the latter being greater. However, the percentage of students who had a distanced classified as “acceptable” was very similar between both genders. When comparing V̇O_2_max, in relative and absolute terms, differences were found, with males obtaining higher values over females. Also, the percentage of the V̇O_2_max was classified as higher in males, and the percentage that was classified as very high was higher in females (Table 2).

Percentile values were developed for the distance traveled in the SMWT and for the CRF expressed in V̇O_2_max in relative terms for the adolescents in the present study according to gender and age (Table 3 and Table 4).

### 3.3. Cardiorespiratory Fitness and Association with Sociodemographic Variables 

Regarding the prevalence of V̇O_2_max in Course Navette, there were significant differences between males and females (x^2^ = 77.497; *p* < 0.001), between WC categories (normal, abdominal obesity, risk of obesity) (x^2^ = 39.065; *p* < 0.001), and between higher and lower cardiometabodic risk according to WHR (x^2^ = 14.518; *p* < 0.001).

It was found that the students who presented a higher cardiometabolic risk according to their WHR had a 66% lower chance of a relative V̇O_2_max classified as high or very high (OR: 0.34 [95% IC: 0.13; 0.76], *p* = 0.005), compared to those with lower cardiometabolic risk. Students who had abdominal obesity or risk of suffering from it had 57% lower chance of having a high or very high V̇O_2_max (ml·kg·min^−1^) (OR: 0.43 [95% IC: 0.27; 0.68], *p* < 0.001), compared to those with WC classified as normal, and overweight or obese students had a 60% lower probability of having a high or very high V̇O_2_max compared to those with a normal of lower BMI (OR: 0.4 [95% IC: 0.28; 0.56], *p* < 0.001). These two V̇O_2_max categories could be protective factors of suffering a higher cardiometabolic risk, abdominal obesity or risk of having it, and of being overweight or obese (Figure 1).

Finally, females were shown to have a 49% lower probability of high or very high V̇O_2_max compared to males (OR: 0.51 [95% IC: 0.37; 0.71], *p* < 0.001).

No significant differences were found in the prevalence of distance covered in the SMWT between males and females (x^2^ = 0.001; *p* = 0.975), or between higher or lower cardiometabolic risk according WHR (x^2^ = 2.141; *p* = 0.143). However, the prevalence of distance walked was different between the three WC categories (normal, abdominal obesity, and risk of abdominal obesity) (x^2^ = 9.588; *p* = 0.008).

Regarding the distance traveled in the SMWT, the students with abdominal obesity or risk of having it had a 46% lower probability of having an acceptable distance (OR: 0.54 [95% IC: 0.36; 0.81], *p* = 0.002), compared to those with a normal waist. Also, those who had a BMI of overweight or obese had a 45% less chance of an acceptable distance (OR: 0.55 [95% IC: 0.4; 0.76], *p* = 0.0002) compared to students with normal or lower BMI. Thus, having an acceptable distance could be a variable that protects against abdominal obesity and being overweight or obese.

On the other hand, female students were less likely of having an acceptable distance in the SMWT compared to male students (OR: 1.0 [95% IC: 0.73; 1.37], *p* = 0.975), and neither did the students who presented a higher cardiometabolic risk according to WHR (OR: 0.62 [95% IC: 0.32; 1.24], *p* = 0.143), regarding those with a lower cardiometabolic risk (Figure 2).

Finally, the students who had an acceptable distance traveled in the SMWT where 186% were more likely to have a high or very high V̇O_2_max (m·kg·min^−1^), compared to the distance traveled classified as “needs improvement” (OR: 2.86 [95% IC: 2.01; 4.11], *p* < 0.001) (Figure 1). 

After analyzing the independent variables potential V̇O_2_max (L·min^−1^) predictability, the best model was: 7.21133 + (0.24301 × Gender) + (-0.00257 × HRr) + (3.97109 × Height) + (0.00148 × Distance) + (4.52351 × WHR).(1)

This equation was obtained with a sample of 597 students (225 male and 371 female) with a value of r = 0.8, R^2^ = 0.64, and an estimation error = 0.38 L·min^−1^ (*p* < 0.001), where the numerical value of the formula according to gender for male was =2 and for female was =1, the heart rate recovery (HRr) was in beats·min^−1^, the height (with two decimals) and distance traveled in SMWT were both in meters, and WHR was written with two decimals (Table 5).

The equation presented validity due to distribution assumption verifications. The independence waste test by Durbin–Watson showed that these were not associated (DW = 1.9123; *p* = 0.1306), the assumption of normality was satisfactorily verified with the Kolmogorov–Smirnov test (D = 0.0251; *p* = 0.4753), and the homoscedasticity was verified graphically, which indicated a similar dispersion (r = 0.00; *p* = 1.0).

Meanwhile, the Bland–Altman diagram did not report differences between V̇O_2_max criterion in Course Navette and the one calculated in the proposed equation (*p* = 0.766), and only 4.5% (27 pairs) of cases were located outside the agreement limits (Figure 3). The mean difference was 0.0047 LO_2_·min^−1^ (95% IC: −0.0262; 0.0356), being very close to zero; thus, both methods (Course Navette and equation) produced very similar results.

## 4. Discussion

The aim of the present investigation was to generate reference values for the distance traveled in the SMWT and CRF in Course Navette in Chilean adolescents, as well as to generate an equation to estimate the V̇O_2_max. Thus, the main findings of this study were that the 50th percentile of the distance in males was from 607 to 690 m and in females was from 630 to 641 for the age range of 13 to 17 years old in both groups. On the one hand, the CRF expressed in the V̇O_2_max was from 43.9 to 45 and from 37.5 to 31.5 mlO_2_·kg·min^−1^ in males and females, respectively, in the same age range indicated above. Finally, the V̇O_2_max was predicted based on the distance traveled in the SMWT and on a demographic variable such as gender, physiological variable such as heart rate, and basic anthropometry such as height and WHR.

Regarding the findings described above, the comparative evidence indicates that normative values for the SMWT have been established according to the performance of different groups of adolescents in countries of different economic incomes [14,20,21,22,23], where it was noted that the distances traveled vary over a wide range, findings values over 600 and 700 m, which are generally greater in men. However, in Chile, efforts have been made to establish reference data for the distance traveled for children and adolescents, where for the age of 14 years, 638 and 674 m were reported as an average value for females and males, respectively [24], with this being the maximum age investigated. In this study, considering that the average age for both genders was 15 years old, the 50th percentile value indicated 643 m for females and 730 for males, which could suggest that the latter have had an increase in their physical fitness over the years compared to the study by Gatica et al. [24].

Regarding CRF, a recently published study reports values for adolescents from countries of high to low income including countries from various continents and from Latin American and the Caribbean, including Chile, demonstrating that the 50th percentile for male and female age 15 is 44 and 37 mlO_2_·kg·min^−1^ respectively [1]. It was found that Chilean adolescents had a somewhat similar trend since it reported values of 41 and 34 mlO_2_·kg·min^−1^ according to gender and the same chronological age, showing that the fitness level should improve in these adolescents. It is relevant that these individuals have an adequate level of CRF since it had been shown that it can be used to predict cardiometabolic risk in adolescents [25], which has been shown to have inverse relationships with body adiposity [26,27] and also with metabolic variables in children and adolescents, so the development of CRF at an early age can be related to good health in adulthood [28,29].

Regarding the models that predict the CRF through the performance in the SMWT and variables related to health indicators, positive relationships have been found between the distance in the SMWT and the V̇O_2_max in apparently healthy children and adolescents and those with some pathologies [30,31,32,33,34]. Therefore, formulas have been developed that consider, in addition to the distance traveled, aspects of body adiposity such as BMI [35,36] in children and adolescents, and other demographic variables incorporating basic anthropometry, physical activity level, and heart rate in children with pathologies [37]. In addition to gender and basic anthropometry, the independent variables that predict CRF in the proposed model of this study incorporated heart rate obtained after performing SMWT, which confirms that it is a novel aspect that is easy to measure and that it has been used in other research and enhances the idea of the practical utility that it may have in its involvement with cardiovascular health during and after physical exertion.

This investigation was not without limitations; one of them is that the criterion test to determine V̇O_2_max was through an indirect method, so it would be desirable that future research measures it directly in a stress with ergospirometry. Given this, progress has been made with the measurement of a “gold standard” for a V̇O_2_max prediction model in SMWT in the young population [38]. However, when there is no access to an ergospirometry test due to the limitations that this implies (economic time, trained personal, etc.), an alternative is testing lung capacity and volume, which have shown significant correlations with V̇O_2_max in children and adolescents [39], but their association with SMWT and CRF is unknown.

Moreover, the strength of the research was that it worked with a large number of participants, which enhanced statistical analysis and the results themselves, and that accessible means were used for data collection, which are tools for daily use by physical education teachers and physical activity and health professionals, which shows the great relevance of being able to make these means transferable for research from the practical field.

## 5. Conclusions

It is concluded that there were differences between both genders in the distance travelled in the walking test, and for cardiorespiratory fitness in relative and absolute values, with those being greater in males. Also, the references values established for the SMWT and CRF (V̇O_2_max) can guide the established performance standards in Chilean adolescents according to gender and age. It was also concluded that it was possible to predict V̇O_2_max for SMWT according to the performance on the same test, according to gender, and with independent variables related to body morphology. These two tools can be used to model the physical fitness of Chilean adolescents, considering geographical, environmental, cultural, and genetic differences when implementing them.

In additions to the utility of the results of this study, it is desirable that competent institutions such as the Ministry of Sports or the Ministry of Education generate public policies for the development and assessment of physical fitness in the school adolescent population in Chile.

## Figures and Tables

**Figure 1 ijerph-18-02474-f001:**
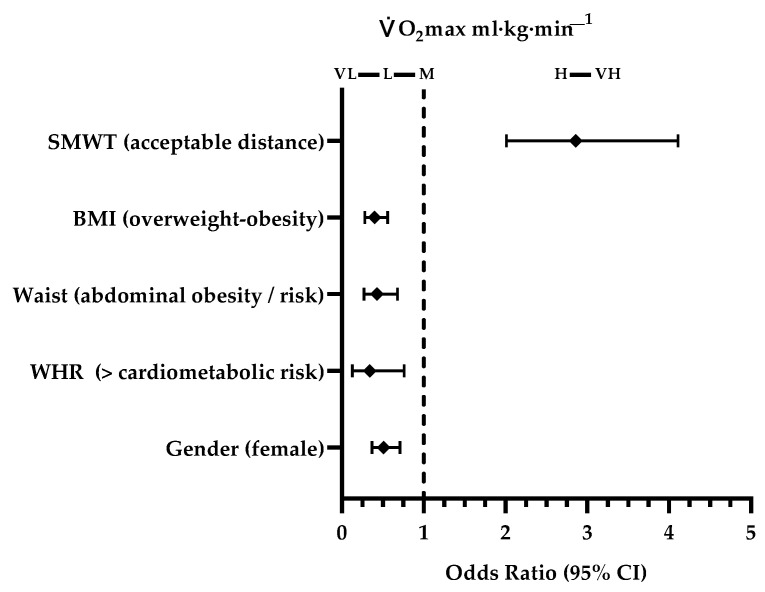
Association between the V̇O_2_max, gender, basic anthropometry, and SMWT.

**Figure 2 ijerph-18-02474-f002:**
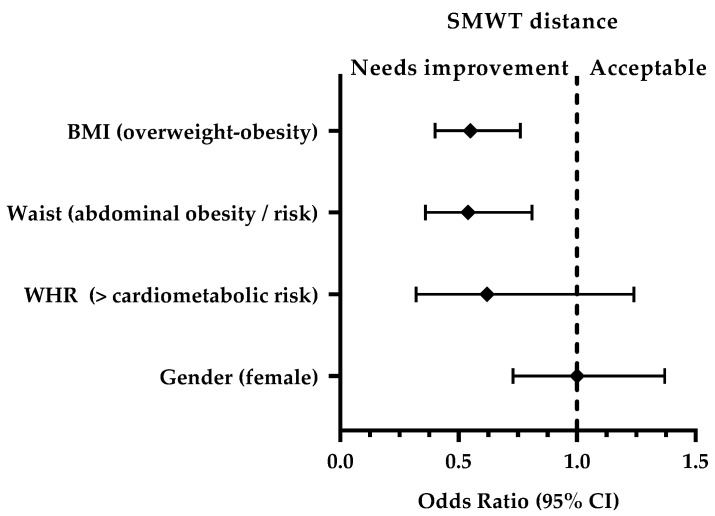
Association between distance traveled (SMWT), gender, and basic anthropometry.

**Figure 3 ijerph-18-02474-f003:**
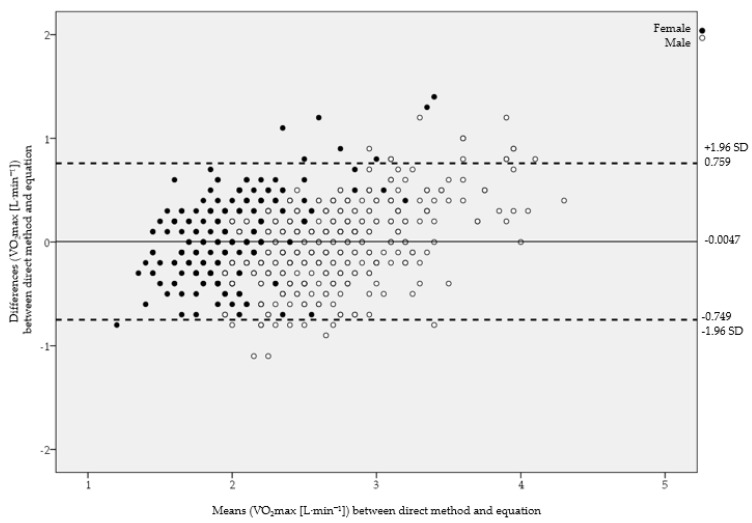
Bland–Altman diagram: agreement between criteria method and equation.

**Table 1 ijerph-18-02474-t001:** Basic anthropometry of adolescents.

	Total (741)		Male (330)		Female (411)		*p*-Value ^a^
Variables	Mean SD	CI	Mean SD	CI	Mean SD	CI	
Age (years)	15.7 (1.1)	(15.7; 15.8)	15.7 (1.1)	(15.6; 15.9)	15.8 (1.1)	(15.6; 15.9)	0.61t
Weight (kg)	62.7 (12.4)	(61.8; 63.6)	65.7 (13.6)	(64.2; 67.1)	60.3 (10.9)	(59.3; 61.4)	<0.001u
Height (m)	1.64 (0.08)	(1.63; 1.65)	1.70 (0.06)	(1.69; 1.71)	1.59 (0.06)	(1.59; 1.6)	<0.001u
BMI (kg·m^−2^)	23.1 (3.9)	(22.8; 23.4)	22.5 (4)	(22.1; 23)	23.5 (3.8)	(23.2; 23.9)	<0.001u
Waist (cm)	76.1 (9.7)	(75.2; 76.8)	78.5 (10.6)	(77.2; 79.8)	74 (8.4)	(73.1; 75)	<0.001u
Normal (n, %)	459 (76.9)	(73.2; 80.2)	187 (70.3)	(64.4; 75.7)	272 (82.2)	(77.6; 86.1)	
Abdominal obesity (n, %)	30 (5)	(3.4; 7.1)	21 (7.9)	(4.9; 11.8)	9 (2.7)	(1.2; 5)	
Obesity risk ab (n, %)	108 (18.1)	(15.1; 21.4)	58 (21.8)	(16.9; 27.2)	50 (15.1)	(11.4; 19.4)	
Total (n, %)	597 (100)		266 (100)		331 (100)		
WHR	0.46 (0.05)	(0.46; 0.47)	0.45 (0.06)	(0.45; 0.47)	0.46 (0.05)	(0.46; 0.47)	0.09u
<CM risk (n, %)	554 (92.8)	(90.4; 94.7)	244 (91.7)	(87.7; 94.7)	310 (93.7)	(90.4; 96)	
>CM risk (n, %)	43 (7.2)	(5.2; 9.5)	22 (8.3)	(5.2; 12.2)	21 (6.3)	(3.9; 9.5)	
Total (n, %)	597 (100)		266 (100)		331 (100)		

^a^: difference between male and female; ab: abdominal; BMI: body mass index; CI: confidence interval; CM: cardiometabolic (risk); SD: standard deviation; t: T-Student test; u: Mann–Whitney U test; WHR: waist-to-height ratio.

**Table 2 ijerph-18-02474-t002:** Cardiorespiratory fitness in Course Navette and SMWT.

	Total (741)		Male (330)		Female (411)		*p*-Value ^a^
Variables	Mena SD	CI	Mena SD	CI	Mena SD	CI	
SMWT							
Distance (m)	668.2 (82.7)	(662.2; 674.1)	699.8 (84.8)	(690.6; 709)	642.8 (71.7)	(635.8; 649.7)	<0.001u
HR (beats·min^−1^)	138 (27)	(136; 140)	138 (27)	(135.2; 141)	138 (28)	(135.6; 141)	0.985u
HR (%)	67.7 (13.6)	(66.7; 68.7)	67.6 (13.3)	(66.2; 69.1)	67.7 (13.8)	(66.4; 69.1)	0.994u
RPE	3.7 (1.7) *	(3.6; 3.8)	3.6 (1.8) **	(3.4; 3.8)	3.8 (1.6) ***	(3.7; 4)	0.039u
Distance (category)							
Acceptable (n, %)	472 (63.7)	(0.6; 0.67)	210 (63.6)	(58.1; 68.8)	262 (63.8)	(58.8; 68.4)	
Needs improvement (n, %)	269 (36.3)	(0.32; 0.39)	120 (36.4)	(31.1; 41.8)	149 (36.2)	(31.5; 41.1)	
Course Navette							
Bearing	4.8 (2.1)	(4.6; 4.9)	6.1 (2)	(5.8; 6.3)	3.7 (1.5)	(3.6; 3.9)	<0.001u
Speed (km·h^−1^)	10.3 (1.2)	(10.2; 10.4)	11 (1.)	(10.9; 11.2)	9.7 (1)	(9.6; 9.8)	<0.001u
V̇O_2_max (ml·kg·min^−1^)	38.3 (7.1)	(37.8; 38.8)	42.5 (6.4)	(41.8; 43.2)	35 (5.7)	(34.4; 35.5)	<0.001u
V̇O_2_max (L·min^−1^)	2.4 (0.6)	(2.3; 2.4)	2.7 (0.6)	(2.7; 2.8)	2.1 (0.4)	(2; 2.1)	<0.001u
RPE	8 (1.6) +	(7.8; 8.1)	8 (1.5) ++	(7.8; 8.3)	7.9 (1.7) +++	(7.7; 8.1)	0.57u
V̇O_2_max (category) ml·kg·min^−1^							
Very low (n, %)	166 (22.4)	(0.19; 0.25)	78 (23.6)	(19.1; 28.5)	88 (21.4)	(17.5; 25.6)	
Low (n, %)	291 (39.3)	(0.35; 0.42)	88 (26.6)	(21.9; 31.7)	203 (49.4)	(44.4; 54.3)	
Moderate (n, %)	18 (2.4)	(1.4; 3.8)	18 (5.5)	(3.2; 8.4)	0	0	
High (n, %)	202 (27.3)	(24; 30)	126 (38.2)	(32.9; 43.6)	76 (18.5)	(14.8; 22.5)	
Very high (n, %)	64 (8.6)	(6.7; 10.8)	20 (6.1)	(3.7; 9.2)	44 (10.7)	(7.8; 14.1)	

^a^: difference between male and female; CI: confidence interval; HR: heart rate; RPE: rated perceived exertion; SD: standard deviation; SMWT: six-minute walk test; t: T-Student test; u: Mann–Whitney U test; V̇O_2_max: maximum oxygen consumption. * n = 675; ** n = 296; *** n = 379; + n = 516; ++ n = 216; +++ n = 300.

**Table 3 ijerph-18-02474-t003:** Percentiles of distance traveled in the SMWT by female and male according to age.

Age	p10	CI	p20	CI	p30	CI	p40	CI	p50	CI	p60	CI	p70	CI	p80	CI	p90	CI
Female																		
13	430	390; 613	578	390; 623	612	418; 630	621	519; 666	630	604; 686	642	617; 726	676	622; 760	700	630; 768	757	673; 768
14	551	522; 580	600	570; 610	612	600; 630	630	612; 640	644	630; 660	660	648; 670	675	660; 690	690	678; 700	707	690; 761
15	570	541; 600	610	600; 621	622	611; 630	630	624; 640	643	630; 660	660	645; 690	690	660; 700	705	690; 720	720	714; 756
16	552	540; 580	590	566; 609	609	592; 615	616	609; 639	639	615; 660	660	638; 672	672	660; 690	690	673; 711	720	700; 755
17	549	506; 570	594	562; 610	610	600; 627	627	610; 642	641	627; 660	660	640; 680	680	660; 696	698	680; 716	720	710; 769
Male																		
13	497	480; 585	576	480; 600	586	503; 610	598	572; 653	607	585; 744	620	591; 759	730	603; 785	756	610; 880	810	740; 880
14	600	585; 620	631	602; 651	658	636; 675	675	658; 695	690	675; 720	720	690; 646	745	720; 780	780	750; 810	826	802; 867
15	630	609; 656	660	637; 690	690	660; 702	702	690; 730	730	700; 750	750	730; 758	758	750; 786	790	763; 810	836	800; 916
16	582	560; 614	630	590; 641	644	630; 670	670	643; 690	690	670; 696	695	689; 716	715	695; 727	729	716; 750	759	730; 781
17	605	542; 633	650	612; 665	664	645; 680	678	661; 690	690	674; 707	700	690; 745	735	699; 770	770	731; 786	790	770; 800

CI: confidence interval; p: percentile.

**Table 4 ijerph-18-02474-t004:** Percentiles of V̇O_2_max (ml·kg·min^−1^) of Course Navette in female and male according to age.

Age	p10	CI	p20	CI	p30	CI	p40	CI	p50	CI	p60	CI	p70	CI	p80	CI	p90	CI
Female																		
13	30.8	30.8; 37.4	36.1	30.8; 37.5	37.4	30.8; 37.6	37.5	32.6; 37.7	37.5	37.4; 38.2	37.6	37.5; 42.6	37.7	37.5; 44.1	39.6	37.6; 44.2	44.1	37.4; 44.2
14	29.3	29.2; 29.8	30.1	29.6; 35.5	35.7	30.3; 35.9	35.9	35.7; 36.2	36.2	35.9; 36.3	36.3	36.2; 36.7	36.8	36.3; 37.2	37.2	36.8; 42.7	43.1	42.4; 43.6
15	27.6	27.4; 27.8	28.3	27.9; 33.8	33.9	28.4; 34.5	34.5	34; 34.7	34.8	34.5; 35.2	35.2	34.9; 35.4	35.4	35.2; 40.9	41.1	40.7; 41.6	41.8	41.3; 45.7
16	25.7	25.2; 26.5	26.7	26; 32.3	32.3	28.7; 32.9	32.9	32.3; 33.2	33.2	32.9; 39	37.9	33.2; 39.4	39.4	38.1; 39.8	40	39.4; 41.1	46	40.3; 46.6
17	24	23.8; 30.5	30.6	24.3; 31	31	30.7; 31.2	31.2	31; 31.5	31.5	31.2; 31.7	31.6	31.5; 37.9	37.9	31.6; 38.4	38.5	37.9; 38.6	38.6	38.5; 38.8
Male																		
13	30.9	30.8; 37.5	33.5	30.8; 43.7	37.5	30.8; 43.9	42.5	31.8; 44.1	43.9	37.4; 47	44	38.6; 50.4	44.1	43.9; 50.5	50.3	44; 50.6	50.5	45; 50.6
14	35.8	35.5; 36.1	36.3	35.9; 36.2	36.6	36.3; 42.4	42.4	36.7; 42.9	42.9	42.3; 43.5	43.4	42.9; 49.1	49.1	43.4; 49.3	49.3	49.2; 49.6	49.7	49.4; 49.9
15	35	34.5; 40.6	40.7	35.2; 41	41.1	40.7; 41.4	41.4	41.1; 42	41.9	41.4; 47.6	47.6	41.9; 48	48	47.6; 48.3	48.3	48; 48.4	48.4	48.3; 49.7
16	32.7	26.1; 33.3	33.4	32.9; 39.1	39.1	33.4; 39.7	39.7	39.1; 40.1	40.1	39.7; 46.3	46.3	40.1; 46.5	46.5	46.3; 46.9	46.9	46.5; 47.3	47.3	47; 53.6
17	31.4	30.9; 37.9	37.9	31.4; 38.4	38.4	37.9; 39.2	38.8	38.3; 45	45	38.7; 45.3	45.2	44.9; 45.5	45.5	45.2; 45.6	45.6	45.5; 45.9	49	45.5; 52.5

CI: confidence interval; p: percentile.

**Table 5 ijerph-18-02474-t005:** Model that explains V̇O_2_max (L·min^−1^).

	Non-Standardized Coefficients				95% CI for B	
	B	Standard Error	t	*p*-Value	Lower Limit	UPPER LIMIT
Constant	−7.21133	0.4261	−16.921	<0.001	−8.0483	−6.3743
Gender	0.24301	0.0427	5.684	<0.001	0.159	0.3269
HRr	−0.00257	0.0005	−4.555	<0.001	−0.0036	−0.0014
Height	3.97109	0.2574	15.424	<0.001	3.4654	4.4767
Distance	0.00148	0.0002	7.054	<0.001	0.0011	0.0018
WHR	4.52351	0.2912	15.533	<0.001	3.9515	5.0954

WHR: waist-to-height ratio; HRr: recovery heart rate.

## Data Availability

The data presented in this study are available on request from the corresponding author. The data are not publicly available due to privacy and anonymity.

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
