# Peer review of "Cardiorespiratory Fitness: Reference on the Six-Minute Walk Test and Oxygen Consumption in Adolescents from South-Central Chile"

_ijerph, 2021, doi:10.3390/ijerph18052474_

Round 1

Reviewer 1 Report

The authors want their results to serve as a reference for future studies.

The population of this study is from the south of Chile, we advise recording the origin of the sample in the title and urge repeating the study in other parts of the country in order to have a representative sample of all Chilean adolescents and, in this way, to verify the trend expressed by the results, which do not show significant differences.

We think conclusions can be improved

Reviewer 2 Report

Hello.

The article presents a research that meets the conditions of publication at a high level.
However, I ask for some clarification:

- what are the values, in meters, in which you classified the categories "acceptable" or "needs improvement" (82), as well as those in the Course Navette test? (90)

- what are the performance standards? Performance in / at what? (312)

- who performed the identified heart rate in the SWMT test? (80)

  • if it was performed by the student, do you consider that determining the heart rate by palpating the carotid is a scientific method?

Thank you.

All the best.

Reviewer 3 Report

1) How can explain the authors the assumption that the physical activity in Chilean adolescents has been reduced over time (Gática et al, 2012), however male (15 years old; 730) but not female (13-17 years old) adolescents increased your physical fitness? The study of Gatica and colleagues shows that this parameter is increased regarding age (6-14 años). Indeed, table 3 of the present work shows that this behavior is replicated in female adolescents from 13-15 years old and maintained until 17 years old. Although, in 16-17 years old males this value corresponds to 690 a 5.5% lesser than 15 years old adolescents.

2) How many adolescents (741) from 13, 14, 15, 16, and 17 years old were included in the present work? Could the age change some data (table 2) obtained in the present work?
